

# Evaluation of film stimuli for the assessment of social-emotional processing: a pilot study

Jenni Leppanen[1], Olivia Patsalos[2], Sophie Surguladze[2], Jess Kerr-Gaffney[2], Steven Williams[1] and Ketevan Tchanturia[2,3,4]

[1] Department of Neuroimaging, King's College London, University of London, London, United Kingdom
[2] Department of Psychological Medicine, King's College London, University of London, London, United Kingdom
[3] South London and Maudsley NHS Foundation Trust National Eating Disorder Service, London, United Kingdom
[4] Psychology Department, Illia State University, Tbilisi, Georgia

## ABSTRACT

**Background**. Difficulties in top-down and bottom-up emotion generation have been proposed to play a key role in the progression of psychiatric disorders. The aim of the current study was to develop more ecologically valid measures of top-down interpretation biases and bottom-up evoked emotional responses.

**Methods**. A total of 124 healthy female participants aged 18–25 took part in the study. We evaluated two sets of 18 brief film clips. The first set of film clips presented ambiguous social situations designed to examine interpretation biases. Participants provided written interpretations of each ambiguous film clip which were subjected to sentiment analysis. We compared the films in terms of the valence of participants interpretations. The second set of film clips presented neutral and emotionally provoking social scenarios designed to elicit subjective and facial emotional responses. While viewing these film clips participants mood ratings and facial affect were recorded and analysed using exploratory factor analyses.

**Results**. Most of the 18 ambiguous film clips were interpreted in the expected manner while still retaining some ambiguity. However, participants were more attuned to the negative cues in the ambiguous film clips and three film clips were identified as unambiguous. These films clips were deemed unsuitable for assessing interpretation bias. The exploratory factor analyses of participants' mood ratings and evoked facial affect showed that the positive and negative emotionally provoking film clips formed their own factors as expected. However, there was substantial cross-loading of the neutral film clips when participants' facial expression data was analysed.

**Discussion**. A subset of the film clips from the two tasks could be used to assess top-down interpretation biases and bottom-up evoked emotional responses. Ambiguous negatively valenced film clips should have more subtle negative cues to avoid ceiling effects and to ensure there is enough room for interpretation.

Corresponding author
Jenni Leppanen,
jenni.leppanen@kcl.ac.uk

## INTRODUCTION

Emotional responses have been proposed to be generated through two processes: top-down and bottom-up (*McRae et al., 2012*; *Ochsner et al., 2009*). Although everyday emotional experiences are likely to arise from a blend of these two processes, top-down and bottom-up emotion generation have been proposed to utilise distinct psychological pathways (*McRae et al., 2012*; *Ochsner et al., 2009*). Top-down emotion generation involves cognitive appraisal of a situation and is influenced by the person's thinking style and past experiences. For instance, due to excessive focus on negative cues or overgeneralisation of previous unpleasant experiences, a person can end up developing a general tendency to interpret neutral, ambiguous situations as negative resulting in negative emotional responses and low mood (*Davis, Foland-Ross & Gotlib, 2018*; *McRae et al., 2012*). Bottom-up emotion generation, on the other hand, refers to reactions arising from exposure to emotionally provoking stimuli rather than from cognitive appraisal (*McRae et al., 2012*; *Ochsner et al., 2009*). As such bottom-up emotion generation is less influenced by cognitive biases, but rather an inherent or habitual reaction with little conscious processing. These include biologically prepared fear responses to seeing a predator and flashbacks following exposure to trauma related places or sounds (*Guillery-Girard et al., 2013*; *McRae et al., 2012*).

The top-down emotion generation pathway is often linked to interpretation biases, which can interfere with the way a person perceives various situations and influence subjective emotional experiences (*Davis, Foland-Ross & Gotlib, 2018*; *McRae et al., 2012*). Over the years, a variety of methods have been used to assess interpretation biases ranging from disambiguating homophones to rating the valence of ambiguous video clips (*Schoth & Liossi, 2017*). Regardless of the exact method used, stimuli evaluation is essential to reliably assess interpretation bias (*Schoth & Liossi, 2017*). It is important to ensure that the stimuli used is ambiguous so that different interpretations are possible, but evaluation of any dominance and unbalance in the possible interpretations is also crucial. For instance, in a homophone paradigm, if participants are presented with spoken unbalanced homophones such as "gilt", and then asked to write what they just heard. In this scenario a participant would likely give a negative answer and write "guilt", rather than providing the neutral answer, "gilt". This could create the impression of negative bias, even though the response bias is more likely to be driven by word dominance. Additionally, dominance effects can result in ceiling effects which make it difficult to examine group differences in interpretation bias in a case-control paradigm.

Although interpretation biases have been extensively studied over the years, recent reviews have reported substantial between study variability which was at least in part explained by methodology (*Everaert, IR & Koster, 2017*; *Chen, Short & Kemps, 2020*). One meta-analysis reported that significant effects were found only when direct measures, such as the sentence completion paradigm and identification of ambiguous facial expressions, were used (*Everaert, IR & Koster, 2017*). However, even amongst studies employing direct measures, heterogeneity was substantial. Another meta-analysis found significant differences between studies employing pictures or videos and those that used written or spoken stimuli only (*Chen, Short & Kemps, 2020*). These findings suggest that, in addition

of dominance effects, the type of stimuli can also influence the results. This may be linked to differences in ecological validity. Indeed, over the years questions have been raised about the real-life generalisability of simple tasks, such as emotion recognition tasks, using still images presented without further context (*Greenaway, Kalokerinos & Williams, 2018*; *Hogenelst, Schoevers & Rot, 2015*). Performance on these types of tasks has not been found to reflect day-to-day social functioning (*Janssens et al., 2012*). Similarly, sentence completion paradigms, which involve reading or listening to a sentence that is missing an ending and then providing a short one or two word ending for the sentence, have been criticised for not reflecting complexities of everyday experiences and for lacking context and visual cues (*Jurado & Rosselli, 2007*; *Chan et al., 2008*). Taken together, there is need to develop more naturalistic paradigms that reliably assess interpretation bias.

The use of film stimuli could offer an easy to implement way of overcoming some of above-mentioned limitations. Previous reviews of studies examining interpretation bias have suggested that using film clips rather than still images to depict ambiguous social situations could help increase vividness of the stimuli and thus increase ecological validity (*Schoth & Liossi, 2017*; *Chen, Short & Kemps, 2020*). Indeed, films can be used to present naturalistic, dynamic social interactions, which can provide a reasonable approximation of real-life situations (*Sonkusare, Breakspear & Guo, 2019*; *Schaefer et al., 2010*). A recent study examining consensus in appraisal of film stimuli documented low degree of agreement between participants indicating that film stimuli may be an effective way to examine highly personal interpretations (*Wallisch & Whritner, 2017*). Additionally, tendency to appraise neutral film clips negatively has also been reported to be strongly associated with emotional responses to daily life events among people with depression (*Panaite, Whittington & Hindash, 2018*). Taken together, these findings suggest that evaluation of ambiguous film stimuli to effectively and reliably study appraisal tendencies or interpretations biases is of interest.

Unlike top-down emotion generation, bottom-up emotions tend to be generated through immediate reactions to the stimuli presented and they are believed to be less influenced by cognitive processes (*McRae et al., 2012*; *Ochsner et al., 2009*). Several different methods have been developed to assess bottom-up emotional reactivity ranging from imagination to emotionally provoking pictures and film clips (*Siedlecka & Denson, 2019*; *Quigley, Lindquist & Barrett, 2014*). As with interpretation bias, stimuli evaluation is important to reliably examine emotional reactivity. However, unlike interpretation bias, ambiguity should be avoided. If a stimulus is open to interpretation, it is less likely produce in consistent immediate emotional reactions and it could result in blend of bottom-up and top-down emotional responses (*Aguado et al., 2018*; *Sheppes & Gross, 2011*).

Brief film clips are commonly used to study the bottom-up emotion generation pathway. Over the years, film stimuli have been reported to be an effective and powerful tool to reliably provoke emotional responses resulting in several research databases (*Deng, Yang & Zhou, 2017*; *Gilman et al., 2017*; *Samson et al., 2016*; *Schaefer et al., 2010*). However, most of the previous work have only evaluated the film stimuli based on self-reported mood ratings. Incorporating other measures, such as spontaneous facial expressions, could help provide further information about how the stimuli alters a person's emotional experience
and how they communicate such experiences. Facial expressions of emotions have a complex role in social interactions, and they are crucial for communication, connection, and building rapport with others (*Schmidt & Cohn, 2001*; *Parkinson, 2005*). Indeed, recent work evaluating importance of cultural specificity of emotional film clips reported that stimuli depicting the participants' own culture provoked stronger facial affective responses than non-culturally specific stimuli (*Alghowinem et al., 2019*). However, similar effects were not observed in the self-reported emotion ratings (*Alghowinem et al., 2019*). Thus, there is interest in developing paradigms that could be used to assess how evoked facial affect might change as the illness progresses or following an intervention. To achieve these goals, it is important to evaluate emotionally provoking film stimuli not only in terms of self-reported emotional states but also evoked facial expressions.

Although previous studies have used film clips which have been found to effectively and reliably induce intended self-reported emotional responses, most of the stimuli used were from well known movies and television shows (*Deng, Yang & Zhou, 2017*; *Gilman et al., 2017*; *Schaefer et al., 2010*). This can introduce problems as participants can have personal memories attached to the stimuli which can be evoked by watching the film resulting in unpredictable responses (*Conway & Loveday, 2010*; *Clark, Mackay & Holmes, 2013*; *Arnaudova & Hagenaars, 2017*; *Maksimainen et al., 2018*). For instance, personal memories have been found to influence the pleasantness and unpleasantness ratings of emotionally provoking musical and pictorial stimuli more than the features of the stimuli alone (*Maksimainen et al., 2018*). Additionally, those who report more spontaneous positive autobiographical memories following an experimental task were found to report greater positive mood reactions to the film clip used in the task (*Clark, Mackay & Holmes, 2013*). These findings suggest that personal memories can impact participants' reactions to a given task, indicating that developing paradigms that use stimuli that participants are not familiar with would be of interest to avoid these possible confounding effects.

Both top-down and bottom-up emotion generation are relevant in the context of psychiatric disorders. Interpretation bias, particularly negative interpretation bias, has been widely researched in psychiatric disorders, including mood, anxiety, and eating disorders, and negative biases have been associated with low mood and worse psychopathology (*Rowlands et al., 2020*; *Everaert, Podina & Koster, 2017*; *Hirsch et al., 2016*). Similarly, alterations in bottom-up emotional reactivity, particularly reduced facial expression of emotions, have been documented in mood and eating disorders (*e.g.*, *Panaite, Whittington & Hindash, 2018*; *Davies et al., 2016*; *Leppanen et al., 2017*) and these alterations in bottom-up emotion generation are believed to contribute to illness progression (*Gross, 2002*; *Butler et al., 2003*; *Treasure & Schmidt, 2013*). Even though difficulties in these aspects of top-down and bottom-up emotion generation appear to be transdiagnostic, it has been suggested that it is important to tailor stimuli to the specific psychiatric disorder being investigated as cognitive biases and habitual emotional responses may vary substantially (*Hirsch et al., 2016*). Thus, if stimuli are to be used in a clinical study, it crucial to not only take the above-mentioned methodological points into consideration when selecting the type of stimuli, ensure that the stimuli targets disorder specific processes, but also evaluate the stimuli to ensure reliability of any subsequent case-control findings.

Current pilot study aims to build on previous research by evaluating two sets of film clips with more dynamic expressions of emotion and contextual cues to assess interpretation biases and evoked emotional responses. The film clips were taken from short films that had not received a wide audience in order to avoid impact of personal memories attached to the stimuli and were selected based on themes reported by people with lived experience of eating disorders. First, we aimed to evaluate the reliability of 18 ambiguous film clips. Specifically, we aimed to examine whether these film clips were sufficiently ambiguous so they could be used to assess interpretation bias. We hypothesised that film clips which were initially categorised as neutral, positively valenced, and negatively valenced would mostly elicit interpretations that fell into the corresponding categories, but that other interpretations were also possible, thus demonstrating ambiguity. Second, we aimed to evaluate the reliability of a second set of 18 emotionally provoking film clips. Specifically, we aimed to examine whether the film clips reliably evoked the intended subjective emotional experiences and facial affective responses. We hypothesised that film clips initially categorised as neutral, positive, and negative would evoke the corresponding self-reported mood and facial affect responses. With both stimuli sets, we explored whether any carry-over effects were present or if the present inter-stimulus interval was sufficient. Finally, as top-down and bottom-up emotion generation have been reported to be affected in anxiety, depression, and eating disorders as outlined above, we also explored whether participants interpretations or emotional reactions were associated with relevant psychopathology measures.

## MATERIALS & METHODS

### Participants

Altogether 124 female participants took part in the study. All participants were 18 –25 years old with no current or past mental health problems or neurological diagnoses. The sample size was based on a power calculation for a $X^2$ test of independence conducted using the R package *pwr* (*Champely, 2020*). Participants we recruited through online adverts featured on websites such as http://www.callforparticipants.com/ and amongst King's College London staff and students. Participants met inclusion criteria if they had no current or past mental health or neurological problems, sleep disturbances, or alcohol or drug misuse or abuse. All participants were screened for eligibility using the Structured Clinical Interview for DSM-5 (*First, Williams & Karg, 2015*), which included the non-patient overview and enhanced screening modules. Prior to taking part all participants gave written, informed consent and all study procedures were conducted in accordance with the latest version of the Declaration of Helsinki (2013). Ethical approval was obtained from the King's College London Psychiatry, Nursing and Midwifery Research Ethics sub-committee (ref: HR-19/20-13004).

### Film stimuli

Altogether 36 short film clips were selected for the two tasks: one assessing interpretation of ambiguous film clips and another examining evoked facial expressions and mood in response to emotionally provoking film clips. For further information about the films

evaluated in the present study see Table S1. Prior to inclusion in the evaluation study, all film stimuli were first discussed amongst members of the research team and peer feedback was used to ensure they were appropriate for each task. For the Ambiguous scenarios, we adopted similar methodology used in previous work (*Huppert et al., 2007*; *Cardi et al., 2017*) and used the peer feedback to select 18 film clips with equal number of ambiguous film clips that were either somewhat positively valenced, somewhat negatively valenced, or neutral. To ensure ambiguity for the initial evaluation, we selected only film clips that produced a variety of interpretations and predictions regarding how the situation might end. These peer discussions resulted in selection of film clips such as a video depicting a man and a woman on a date in a restaurant. When the couple are seemingly unexpectedly served a dessert, which they did not order, and the woman insists that the man eats it despite him protesting that he is feeling full. The man then finds something hard inside the dessert resulting in a surprised expression. The film clip ends as the woman begins a speech that sounds like a proposal. Similarly, for the Evoked emotions task, the feedback was used to select a set of 18 film clips with equal number of positive and negative emotionally provoking film clips as well as neutral films. Through peer discussion we ensured that the film clips were not ambiguous, producing a variety of reactions, but rather evoked the intended mood states. These discussions resulted in the selection of film clips such as a video depicting two police officers who were newly assigned as partners. In the beginning it is clear that one of the officers is not happy to have a new colleague, but throughout the video the two officers become better acquitted with each other and develop a friendship. The video ends with the two officers driving whilst singing along to the radio.

The films were chosen based on themes that were identified in a previous qualitative study asking people with lived experience of anorexia nervosa about critical life events (*Leppanen et al., 2021*) to ensure they will cover themes relevant for people with eating disorders and can thus be used in later case-control studies. The positive themes selected for the two tasks included receiving emotional support, positive messages and reminders from friends and loved ones, and feeling confident. The negative themes selected for the two tasks were feeling unsupported and misunderstood, loneliness, and not feeling good enough. We made sure that these themes touched on or were experienced by the main character in the film clips. Care was taken to ensure that films did not feature themes directly related to eating disorders, such as inpatient treatment, which may alienate healthy participants with no personal experience of eating disorders, thus failing to evoke intended emotions in this group. Additionally, such film clips could trigger unintended emotional responses associated with memories of such situations in future case-control studies.

The film clips were taken from short films freely available on online sharing platforms such as YouTube and Vimeo. We chose to use short films over feature length films, because scenes in such films tend to be succinct and often do not require extensive prior knowledge of events that preceded the scene in order to understand what was happening. This was important as confusion about what was going on in the scene could lead to unpredictable results. Additionally, many people tend not to be as familiar with short films as with feature length films. Unfamiliarity with the film clips was important to ensure participants were not able to predict what was going to happen next. Unfamiliarity with the film clips also

ensured that participants did not have memories attached to the stimuli, which could have influenced their responses in unpredictable ways.

## Self-report measures

Participants were asked to complete self-report questionnaires to obtain information regarding age, body mass index (BMI), ethnicity, employment status, level of education, and marital status. Additionally, participants completed the following standardised questionnaires.

The Eating Disorder Examination Questionnaire (EDEQ; *Fairburn & Beglin, 1994*) is a 28-item self-report questionnaire, which assesses the level of eating disorder cognitions and behaviours experienced over the past 28 days. In the present study the EDEQ had good internal consistency (Cronbach's alpha = 0.90, 95% CI [0.87–0.92]).

The Hospital Anxiety and Depression Scale (HADS; *Zigmond & Snaith, 1983*) is a 14-item self-report questionnaire, which assesses the level of anxiety and depressed mood experienced over the past two weeks. In the present study the HADS had good internal consistency (Cronbach's alpha = 0.84, 95% CI [0.80–0.88]).

The Beliefs about Emotions Scale (BES; *Rimes & Chalder, 2010*) is a 12-item self-report questionnaire assessing participants' general beliefs about experiencing and expressing emotions. In the present study the BES had good internal consistency (Cronbach's alpha = 0.88, 95% CI [0.84, 0.91]).

## Procedures

Prior to the SARS-CoV-2 pandemic participants completed the two tasks at King's College London and those who took part after the pandemic completed the study remotely online. In the office participants first gave written consent, filled in a set of self-report questionnaires, and completed the Ambiguous scenarios task followed by the Evoked emotions task. The two tasks were presented using Psychopy (*Peirce et al., 2019*). When taking part in the study remotely participants completed the same steps online during a video conference with one of the researchers. The online versions of the tasks were otherwise identical to the those completed in the office but were hosted on http://gorilla.sc/ (*Anwyl-Irvine et al., 2020*).

## Ambiguous scenarios task

As part of the Ambiguous scenarios task participants were asked to watch 18 short film clips presented in random order. The film clips varied in length lasting 1.1 to 1.6 min and an inter-stimulus interval of 2.0 s was used. After each film clip participants were asked two questions: (1) "What do you think happened in the video?" and (2) "What do you think might happen next?". Participants were instructed to type brief answers to each question explaining how they perceived the situation and how they thought the situation might move forward. This was done to ensure participants did not only describe the events in the scenarios, but also provided their own interpretations. After the task had finished, participants were also asked if they had seen any of the film clips before to avoid memory effects.

The written responses were subjected to sentiment analysis and were coded in terms of their emotional valence into the following categories: positive, neutral, or negative.

The answers to the two questions were coded separately. The rating was done by two independent coders (O.P. and S.S.) who were not familiar with the film stimuli to avoid bias. The coders were instructed to focus on how participants described the interactions between the people in the film clips rather than descriptions of the surroundings. Inter-rater agreement was acceptable (Cohen's Kappa = 0.65). All disagreements were resolved by a third independent coder (J.K-G) and the final codes were then taken forward for further statistical analysis. The Ambiguous scenarios task took participants approximately 45 min to complete.

### Evoked emotions task

During the Evoked emotions task participants were asked to watch 18 short film clips presented in random order. The film clips varied in length lasting 1.5 to 2.1 min and an inter-stimulus interval of 2.0 s was used. After each film participants were asked to rate their mood and level of alertness using the affective slider (*Betella & Verschure, 2016*). The affective slider is a digital visual analogue scale designed to assess mood/pleasure and alertness/arousal. The present study focused on participants' mood/pleasure ratings. This was done to obtain information about how participants rated their own mood in response to the film clips and how well the stimuli captured their attention. During this task participants' faces were also video recorded to obtain information about evoked facial expressions in response to the film stimuli. The video recordings were analysed using a commercially available automated facial affect analysis tool FaceReader version 8.1 (Noldus Information Technologies). The FaceReader uses a 500 key point mesh and deep learning to classify facial expression in each frame of the video into seven emotion categories including neutral. The FaceReader calculated the valence of facial expressions in each frame by subtracting the highest intensity negative emotion from the intensity of 'happy' emotion. Each participant's neutral facial expression recorded at the beginning of the task was used to first calibrate the analysis and sample rate was set to frame rate. Information regarding each participants' mean valence of evoked facial expressions made during each film clip was extracted from the FaceReader output. After the task had finished, participants were also asked if they had seen any of the film clips before to avoid memory effects. The Evoked emotions task took participants approximately 40 min to complete.

### Data analysis

All data analyses were conducted using R version 4.0.3 (*R Core Team, 2020*). In all significance tests, $p < 0.05$ was considered significant and where appropriate adjustments for multiple comparisons were used. As several participants took part in the study from home due to the SARS-CoV-2 pandemic, we compared participants who completed the tasks in person and those who took part from home. There were no significant differences between the two groups. The results are presented in the Tables S2, S3 and S4.

### Ambiguous scenarios task

First, we assessed whether participants interpreted the ambiguous film clips in an expected manner offering more negative interpretation in response to negative films and positive interpretations in response to positive films. This was done by examining differences

between the film categories in the valence of participants' written responses to the two questions after each film clip. We conducted a cumulative link mixed model from the R package *Ordinal* (*Christensen, 2019*) with film category and question as predictors and valence as the ordinal outcome measure. For ease of interpretation, the *Anova.clmm* command was used to obtain analysis of deviance table. In case of significant main effects or interactions, post-hoc comparisons were conducted using *emmeans* (*Russell, 2021*) and *p*-values were adjusted for multiple comparisons using the Tukey method.

As the film clips in the Ambiguous scenarios task should be ambiguous to ensure that it could be used as an alternative to other ambiguous scenarios tasks, we conducted an ambiguity check using a $X^2$ test of independence. This was done by calculating the relative contributions of each film clip to each interpretation category to establish if participants' written responses to any film clip were strongly associated with a particular interpretation category, which would indicate lack of ambiguity.

As the films were presented in randomised order, the presence of possible carry-over effects was examined by comparing the valence of written responses given in response to film clips that were preceded by negative or positive film clips. A cumulative link model was conducted to examine carry-over effects on neutral, positive, and negative film clips, with preceded by negative and preceded by positive and film category as the predictors, and valence of the written responses to the two questions as the ordinal outcome measure. Finally, Kendall's tau correlation matrix was calculated to explore whether the overall mean valence of all interpretations to the two questions participants offered after each ambiguous film clip were correlated with any of the self-report measures. Overall mean valence of all interpretations was used in an attempt to capture participants general tendency to offer positive, neutral or negative interpretations. In the correlation tests the *p*-values were adjusted for multiple comparisons using the Holm method.

## Evoked emotions task

We conducted an exploratory factor analyses to examine the factor structure in participants self-reported mood rating following each emotional and neutral film clip and the valence of participants facial expressions during each film. The factor analyses were conducted using the R package *psych* (*Revelle, 2020*) with weighted least square factor method to minimise the impact of any potential outliers. First, we determined the number of factors in the data matrices using the parallel method which compares the observed data scree plots with those produced using a same sized random data matrix. The factor exploratory factor analyses were then conducted with the predetermined number of factors and *varimax* matrix rotation utilising Kaiser normalisation. We then examined whether the film clips were split into factors in an expected manner, with neutral film clips forming one factor, positive film clips another, and negative film clips forming a third factor. We also explored if the results from the two factor analyses using self-reported mood ratings and valence of facial expressions produced different factor structures and whether participants mood ratings and facial expressions in response to each video correlated with each other using Kendall's tau correlation tests.

As the films were presented in randomised order, the presence of carry-over effects was examined by comparing the self-reported mood and valence of facial expressions in response to film clips that were preceded by negative or positive film clips. Two separate analyses were conducted to examine carry-over effects on neutral, positive, and negative film clips by conducting multivariate linear mixed models, with preceded by negative and preceded by positive as the predictors and self-reported mood, and mean valence of facial expression as the outcome measures. Finally, Kendall's tau correlation matrix was calculated to explore whether the overall mean mood ratings and valence of facial expressions across all films were correlated with any of the self-report measures. Overall mean mood ratings and valence of facial expressions across all film clips was used in an attempt to capture participants general tendency to rate and express positive, neutral, or negative emotions.

## RESULTS

### Sample characteristics

Sample characteristics are presented in Table 1. Overall participants' responses to the self-report questionnaires fell into a non-clinical range, with low incidence of eating disorder cognitions and behaviours, anxiety, and depressed mood. Participants' general beliefs about emotions and quality of life was comparable to previous studies reporting normative data.

### Ambiguous scenarios task

Quality assessment was conducted prior to any data analysis. Data from two participants was excluded due to internet connectivity issues which made it difficult for the participants to follow and understand the story in the film clips. Additionally, due to internet connectivity difficulties, data was missing from three participants and three other participants were unable to answer all questions. One participant had seen one of the film clips before (Film 5). Her written responses to the two questions after this film clip were not included in further analysis. No other participants reported having seen any of the film clips before. Thus, data from 119 participants were included for further analysis, 116 of whom provided complete answers for both of the two questions after each video.

### Valence of interpretations per film category

The valence summary statistics per film category are presented in Table 2. As expected, there was a significant main effect of film category in the cumulative link mixed model ($X^2$ (2) $= 872.92$, $p < 0.001$). The valence of the written answers to the two questions given in response to film clips categorised as positive were more positive than those given in response to film clips categorised as negative ($z = 28.75$, $p < 0.001$) or neutral ($z = 11.86$, $p < 0.001$). Additionally, the valence of the answers to the two questions following film clips that were categorised as negative were more negative than those given after film clips that were categorised as neutral ($z = -18.27$, $p < 0.001$). The cumulative link mixed model also revealed a significant main effect of question ($X^2$ (1) $= 56.59$, $p < 0.001$), such that participants gave more positive responses to the second question asking them to predict

**Table 1  Participant characteristics (N = 124).**

|  | Current sample | Normative data |
|---|---|---|
| Age<br>M (SD); (range) | 22.0 (2.3); (18 –25) | N/A |
| BMI<br>M (SD); (range) | 22.2 (5.4); (17.8 –39.7) | 18.5 –24.9 |
| EDEQ Total<br>M (SD); (range) | 0.6 (0.6); (0.0 –2.46) | <2.5[1] |
| HADS Anxiety<br>M (SD); (range) | 3.3 (2.8); (0.0 –13.0) | 0 –7 |
| HADS Depression<br>M (SD); (range) | 2.2 (2.5); (0.0 –14.0) | 0 –7 |
| BES<br>M (SD); (range) | 26.1 (11.6); (3.0 –62.0) | Mean = 27.9 (SD = 11.3)[2] |
| Employment status<br>N (%) | Full-time student: 70 (56.5%)<br>Full-time employment: 18 (14.5%)<br>Student in part-time employment: 16 (12.9%)<br>Part-time employment: 8 (6.5%)<br>Unemployment: 6 (4.8%)<br>Student in full-time employment: 1 (0.8%)<br>Did not disclose: 5 (4.0%) | N/A |
| Level of education<br>N (%) | A Level, NVQ or equivalent qualification: 47 (37.9%)<br>Undergraduate degree: 40 (32.3%)<br>Postgraduate degree: 28 (22.6%)<br>BTEC or equivalent Diploma: 3 (2.4%)<br>O Level or General Certificate of<br>Secondary education: 1 (0.8%)<br>No qualifications: 1 (0.8%)<br>Did not disclose: 3 (2.4%) | N/A |
| Marital status<br>N (%) | Single: 105 (84.7%)<br>Domestic partnership: 14 (11.3%)<br>Married: 1 (0.8%)<br>In a relationship: 1 (0.8%)<br>Did not disclose: 3 (2.4%) | N/A |
| Ethnicity<br>N (%) | White: 71 (57.3%)<br>Asian: 29 (23.4%)<br>Mixed: 9 (7.3%)<br>Black: 8 (6.5%)<br>Middle Eastern: 4 (3.2%)<br>Did not disclose: 3 (2.4%) | N/A |

Notes.
[1] (*Rø, Reas & Stedal, 2015*) recommended using EDEQ total score above 2.5 as an indicator of clinical significance.
[2] the mean is based on what was reported in the initial validation of the scale among healthy people (*Rimes & Chalder, 2010*).
BMI, body mass index; HADS, Hospital Anxiety and Depression Scale; EDEQ, Eating Disorder Examination Questionnaire; BES, Beliefs about Emotions Scale; NVQ, National Vocational Qualifications; BTEC, Business and Technology Education Council; SD, standard deviation; N/A, not applicable.

what was going to happen next than to the first question asking them what happened in the film clip ($z = 7.56$, $p < 0.001$). There was no significant film category by question interaction ($X^2$ (2) = 3.79, $p = 0.151$).
| Table 2 Valence of written interpretations. | | |
|---|---|---|
| Film category | Question 1 Valence M (SD) | Question 2 Valence M (SD) |
| Neutral | −0.18 (0.58) | −0.07 (0.78) |
| Positive | 0.09 (0.65) | 0.29 (0.80) |
| Negative | −0.66 (0.56) | −0.51 (0.65) |
| Total | −0.25 (0.67) | −0.10 (0.82) |

## Ambiguity check

The $X^2$ test of independence was significant ($X^2 = 1515.9$, $p < 0.001$), suggesting that the film clips and the valence of participants' written response are dependent as would be expected based on the findings above. The relative contributions of each film clip to the model are presented in Fig. 1. There were three film clips that appeared to make substantial contributions to one valence category over others when compared to the other film clips, suggesting they lacked ambiguity. Film 1 was strongly associated with negative responses to question 1, Film 4 was strongly associated with negative responses to question 1, and Film 8 was strongly associated with positive responses to question 2. These relative contribution statistics are supported by histograms showing the number of neutral, positively valenced, and negatively valenced answers to each film (Fig. 2). Additionally, as shown in the Fig. 2, participants primarily interpreted the negatively valenced film clips in a negative way, while such a pattern was not observed in the interpretations of the neutral and positively valenced film clips.

The dots indicate contribution each film clip made to a specific interpretation category. Larger blue dots indicate higher contribution and lower ambiguity with most participants offering the same type of interpretation.

The histograms show the frequency of positive, neutral and negative answers participants offered to the two questions after each film clip. Films 1–6 were categorised as negatively valenced ambiguous film clips, Films 7–12 were categorised as neutral ambiguous film clips, and Films 13–18 were categorised as positively valenced ambiguous film clips.

## Assessment of carry-over effects

The cumulative link mixed model did not reveal significant effects of preceded by a positive film ($X^2$ (1) = 0.137, $p = 0.711$) or a negative film ($X^2$ (1) = 0.24, $p = 0.625$), suggesting that a preceding emotional film did not impact the interpretation of a subsequent film clip. There were also no significant interactions between preceded by positive film and film category ($X^2$ (2) = 0.98, $p = 0.612$) or preceded by negative film and film category ($X^2$ (2) = 7.21, $p = 0.125$) suggesting there were no carry-over effects within film categories.

## Correlations between valence of interpretations and clinical variables

There were no significant correlations between the mean valence of the written answers participants gave to question 1 or question 2 and self-reported eating disorder symptomatology, anxiety, depression, or beliefs about emotions (Table S5).
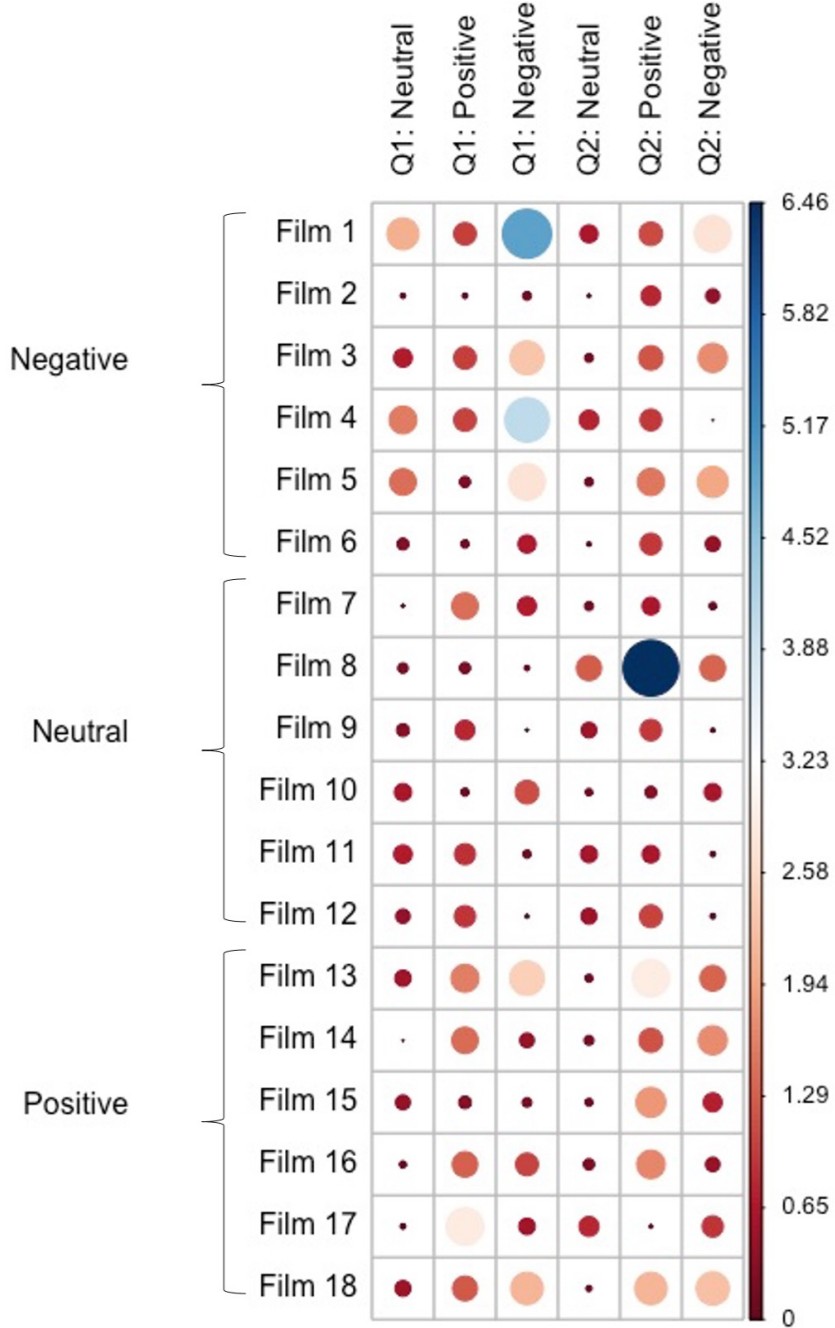

**Figure 1  χ² contribution plot.** The dots indicate contribution each film clip made to a specific interpretation category. Larger blue dots indicate higher contribution and lower ambiguity with most participants offering the same type of interpretation.

## Evoked emotions task

Prior to statistical analysis, data quality was evaluated. As with the Ambiguous scenarios task above, data from two participants was excluded due to internet connectivity issues

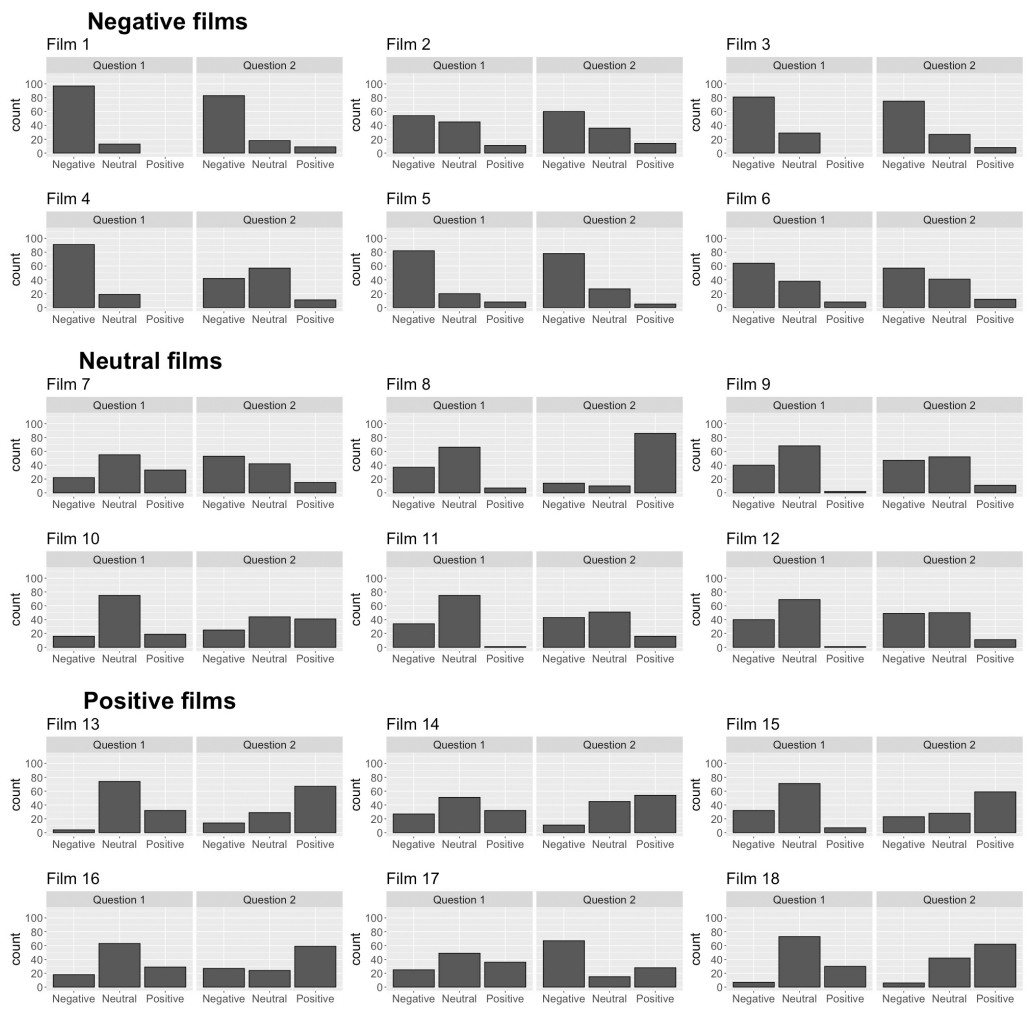

**Figure 2  Evaluation of film stimuli for the assessment of social-emotional processing: A pilot study.**
The histograms show the frequency of positive, neutral and negative answers participants offered to the
two questions after each film clip. Films 1–6 were categorised as negatively valenced ambiguous film clips,
Films 7–12 were categorised as neutral ambiguous film clips, and Films 13–18 were categorised as posi-
tively valenced ambiguous film clips.

which made it difficult for the participants to follow and understand the story in the film
clips. Due to technical difficulties data was missing from one participant and another
two participants were unable to watch all 18 videos during the Evoked emotions task
leading to partially missing data. Thus, self-reported mood ratings were available from
121 participants. Additionally, video recordings from three participants were excluded
because the participants moved outside the camera's range for extended periods of time.
Video recordings of another three participants were excluded from further analysis due
to low quality. Thus, facial expression data was available from 115 participants. Finally,
two participants reported having seem two of the films (Film 5, Film 16) before and their
responses to these films were not included in the final analysis.

**Table 3** Self-rated mood data factor loadings.

| Film | Film category | Factor 1 | Factor 2 | Factor 3 |
|------|---------------|----------|----------|----------|
| Film 6 | Negative | 0.709 | | |
| Film 2 | Negative | 0.683 | | |
| Film 5 | Negative | 0.670 | | |
| Film 4 | Negative | 0.614 | | |
| Film 3 | Negative | 0.449 | | |
| Film 8 | Neutral | 0.531 | | |
| Film 1 | Negative | 0.430 | | |
| Film 12 | Neutral | | 0.819 | |
| Film 11 | Neutral | | 0.498 | |
| Film 10 | Neutral | | 0.428 | 0.406 |
| Film 9 | Neutral | | 0.375 | |
| Film 16 | Positive | | | 0.547 |
| Film 13 | Positive | | | 0.518 |
| Film 18 | Positive | | | 0.511 |
| Film 7 | Neutral | | | 0.402 |
| Film 17 | Positive | | | 0.344 |
| Film 15 | Positive | | | 0.313 |
| Film 14 | Positive | | | |

**Notes.**
Factor loadings <0.3 were excluded.

## Exploratory factor analysis with self-reported mood ratings

It was determined that the self-reported mood rating dataset contained three factors (Fig. S1). Factor 1 comprised of seven film clips and explained 45.1% of the variance with factor loadings ranging from 0.43 to 0.71 (Table 3). Factor 1 included all film clips from the negative category and one neutral film clip. Factor 2 comprised of four film clips, explained 28.9% of the variance and the factor loadings ranged from 0.38 to 0.82. Factor 2 included four of the six neutral film clips with one of the film clips cross-loading with Factor 3. Factor 3 consisted of seven film clips and explained 26.0% of the variance with factor loadings raging from 0.31 to 0.55. Five of the film clips in Factor 3 were from the positive category and two were from the neutral category. Film 14 did not fit in any of the three factors.

The exploratory factor analysis of mood ratings showed that most of the positive, neutral and negative films tended to form their own unique factors, which supports the initial categorisation of the film clips. This is supported by a visual inspection of participants' mood ratings in response to each film clip (Fig. S2).

## Exploratory factor analysis with valence of facial expressions

It was determined that three factors were present in the facial expression data (Fig. S3). Factor 1 consisted of twelve film clips and explained 46.0% of the variance with factor loadings raging from 0.31 to 0.91 (Table 4). Factor 1 included all six film clips from the positive category. The other film clips in Factor 1, five neutral and one negative film clips, cross-loaded with Factors 2 and 3. Factor 2 included eleven film clips and explained 38.2%

**Table 4  Facial expression data factor loadings.**

| Film | Film category | Factor 1 | Factor 2 | Factor 3 |
|------|---------------|----------|----------|----------|
| Film 14 | Positive | 0.912 | | |
| Film 16 | Positive | 0.809 | | |
| Film 15 | Positive | 0.779 | | |
| Film 18 | Positive | 0.761 | | |
| Film 13 | Positive | 0.721 | | |
| Film 10 | Neutral | 0.608 | 0.511 | |
| Film 8 | Neutral | 0.598 | 0.415 | |
| Film 17 | Positive | 0.463 | | |
| Film 1 | Negative | | 0.865 | |
| Film 2 | Negative | | 0.806 | |
| Film 6 | Negative | 0.314 | 0.690 | |
| Film 9 | Neutral | 0.604 | 0.604 | 0.350 |
| Film 3 | Negative | | 0.599 | 0.594 |
| Film 5 | Negative | | 0.589 | 0.337 |
| Film 11 | Neutral | 0.495 | 0.514 | |
| Film 7 | Neutral | | 0.473 | |
| Film 12 | Neutral | 0.397 | | 0.755 |
| Film 4 | Negative | | 0.473 | 0.611 |

**Notes.**
Factor loadings <0.3 were excluded.

of the variance with factor loadings ranging from 0.42 to 0.87. Factor 2 included all six film clips from the negative category, four of which cross-loaded with Factors 3 and 1. Factor 2 also included five of the neutral films, four of which cross-loaded with the other two factors. Factor 3 consisted of five film clips, explained 15.7% of the variance, and the factor loadings ranged from 0.34 to 0.76. Factor 3 included three negative and two neutral film clips, all of which cross-loaded with the other two factors.

The exploratory factor analysis showed that most of the positive film clips appeared to form one factor while most of the negative film clips appeared to form another factor based on participants facial affect while watching the films. The neutral film clips on the other hand tended to cross-load across factors based on participants facial expressions while watching these films. This is supported by a visual inspection of the valence of participants facial expressions during each film clip (Fig. S4).

## Correlation between self-reported mood ratings and evoked facial affect

The exploratory correlation analysis found significant positive correlations between participants mood ratings and the valence of facial expressions produced during most of the positive film clips ($\tau = 0.24 - 0.32$, all $p < 0.05$) apart from two film clips (Film 13: $\tau = 0.21$, $p = 0.05$, Film 17: $\tau = 0.13$, $p = 0.23$). There were no significant correlations were self-reported mood ratings and valence of facial expressions in response to the neutral (all $\tau < 0.14$, all $p > 0.1$) or negative films clips (all $\tau < 0.13$, all $p > 0.1$).

## Assessment of carry-over effects

We assessed whether there were any carry-over effects in participants' self-reported mood ratings or valence of facial expressions. The linear mixed effects model of mood ratings showed a significant interaction between preceding a negative film and film category ($F(4,2063.1) = 523.36$, $p < 0.001$). This interaction was driven by a significant impact of film category such that participants reported more negative mood in response to negative films than neutral ($t(2045) = -23.02$, $p < 0.001$) or positive film clips ($t(2047) = -41.27$, $p < 0.001$) when they were not preceded by another negative film clip. Participants also reported more positive mood when watching the positive than neutral film clips when they were not preceded by a negative film clip ($t(2049) = 17.30$, $p < 0.001$). Similarly, when the film clips were preceded by another negative film clip, participants reported more negative mood while watching the negative film clips than when watching the neutral ($t(2076) = -12.64$, $p < 0.001$) or positive film clips ($t(2077) = -21.69$, $p < 0.001$). Participants also reported more positive mood in response to the positive than neutral film clips ($t(2072) = -9.79$, $p < 0.001$) when they were preceded by a negative film. There was no significant effect of being preceded by another negative film within the negative ($t(2071) = 0.10$, $p = 0.923$), neutral ($t(2067) = 0.17$, $p = 0.866$), or positive film categories ($t(2047) = 0.52$, $p = 0.605$). There also was no significant interaction between preceded by a positive film and film category ($F(2,2087.8) = 0.28$, $p = 0.753$) or significant effects of preceded by a positive film ($F(1,2027.2) = 0.001$, $p = 0.972$) or preceded by a negative film ($F(1,2071.3) = 0.01$, $p = 0.923$) across the film categories. These findings indicate that there were no substantial carry-over effects in participants mood ratings.

As above, the linear mixed effects model of the valence of participants' facial expressions showed a significant interaction of preceded by a negative film and film category ($F(4,1556.0) = 53.67$, $p < 0.001$). This interaction was again driven by a significant effect of film category such that participants expressed more negative emotions in response to negative films than neutral ($t(1559) = -4.60$, $p < 0.001$) or positive film clips ($t(1560) = -13.22$, $p < 0.001$) when the films were not preceded by another negative film. Participants also expressed more positive emotions in response to the positive than neutral films when they were not preceded by a negative film clip ($t(1561) = 8.28$, $p < 0.001$). Similarly, participants expressed more negative emotions in response to negative films than neutral ($t(1566) = -3.62$, $p = 0.001$) or positive film clips ($t(1570) = -6.88$, $p < 0.001$) when the films were preceded by another negative film. Participants also expressed more positive emotions in response to the positive than neutral films when they were preceded by a negative film clip ($t(1568) = 3.56$, $p = 0.001$). There was no significant effect of being preceded by another negative film clip on participants' facial affect when watching negative ($t(1570) = 1.35$, $p = 0.179$), neutral ($t(1565) = 0.06$, $p = 0.952$), or positive films ($t(1565) = 1.61$, $p = 0.108$).

The linear mixed model of facial affect also revealed a similar significant interaction between preceded by a positive film clip and film category ($F(2,1573.8) = 4.12$, $p = 0.016$). As with the other interaction, this was driven by a significant impact of film category such that participants expressed more negative emotions in response to negative films than neutral ($t(1561) = -4.78$, $p < 0.001$) or positive film clips ($t(1558) = -14.96$, $p < 0.001$)

when the films were not preceded by a positive film. Participants also expressed more positive emotions in response to the positive than neutral films when they were not preceded by another positive film clip (t(1560) = 10.18, $p < 0.001$). Similarly, participants expressed more negative emotions in response to negative films than neutral (t(1570) = −3.50, $p = 0.001$) or positive film clips (t(1569) = −5.92, $p < 0.001$) when the films were preceded by a positive film. Participants also expressed more positive emotions in response to the positive than neutral films when they were preceded by another positive film clip (t(1567) = 2.53, $p = 0.031$). There was no significant effect of being preceded by a positive film clip on participants' facial affect when watching negative (t(1568) = −1.06, $p = 0.287$), neutral (t(1571) = −1.94, $p = 0.052$), or positive films (t(1563) = 1.93, $p = 0.054$). There were no significant effects of being preceded by a negative (F(1,157.8) = 1.81, $p = 0.179$) or a positive film clip (F(1,1559.7) = 0.43, $p = 0.513$) across all film categories suggesting no substantial carry-over effects were precent in the facial expression data.

### Correlations between self-reported mood, valence of facial expressions and self-report questionnaire measures

There were no significant correlations between mean mood rating across all films, valence of facial expressions across all films, eating disorder symptomatology as measured by the EDEQ total score, anxiety as measured by HADS, or the BES total score (Table S6).

## DISCUSSION

The present pilot study aimed to conduct initial evaluation to identify film stimuli that could be used in two tasks assessing different aspects of emotion generation, namely top-down interpretation of ambiguous scenarios and bottom-up evoked emotional responses. Most participants were not familiar with the short films used in the present study, which allowed us to avoid memory effects. Most of the film clips in the Ambiguous scenarios task elicited interpretations that fell into the expected neutral, positive, and negative categories. Additionally, with the exception of three film clips, they appeared to also be ambiguous with participants offering different types of interpretations. The exploratory factor analyses revealed that positive and negative film clips similarly formed their own factors when using participants self-reported mood and valence of facial expressions. No significant carry-over effects were observed in either task.

As anticipated participants mostly offered valence-congruent interpretations. Additionally, most film clips appeared to be ambiguous enough to allow participants to offer different interpretations, with only three of the film clips being deemed unambiguous. Although this suggests that most of the film clips would be suitable for use in a more naturalistic version of the sentence completion task to assess interpretation biases, it is important to note that the participants offered primarily negative interpretations for most negatively valenced ambiguous film clips and no or very few positive interpretations. The opposite pattern was not observed in the interpretations of the positively valenced film clips. This suggests that participants were more vigilant to detect negative cues and more subtle negative cues may be needed to preserve ambiguity. Similar issues have been reported in a previous study, where healthy participants' interpretation bias score was negative

even after positive training indicating greater vigilance towards negative scenarios (Van (*Bockstaele et al., 2019*). Another factor at play could be the stimulus duration. A previous study exploring interpretation of negative facial expressions reported that increasing the duration of stimulus presentation appeared to result in increased ambiguity, that is fewer negative and greater number of alternative interpretations (*Vassilopoulos, 2011*). Having more time with the stimulus could give participants more opportunities to engage with cognitive appraisal and think of alternative explanations for the facial expressions. However, it is important to note that the negative faces were only presented for 200 or 500 ms, which may not be directly comparable to film stimuli. To our knowledge, no studies to date have examined the impact of film duration on ambiguity. Future studies may benefit from utilising more subtle negative elements to examine interpretation biases in negatively valenced ambiguous films as well as exploring the impact of stimulus duration.

Interestingly, participants frequently offered answers to question 1 that went beyond neutral description of the scene they were presented with, suggesting that the analysis of participants' responses to the present Ambiguous scenarios task could be expanded. In the future, this task could be used to expand the field of interpretation bias research by separately examining how participants generally view the current situation and how they view the future. Additionally, the present paradigm involves participants writing complete sentences to answer the two questions after each film, a qualitative analysis that goes beyond rating the valence of the answers could also be employed. Using qualitative methods, such as thematic analysis, could enable researchers to examine more subtle differences in the written responses and gain deeper insight into the participants' thinking style. This method has been used to assess participants' responses to the Frith-Happé theory of mind task (*Abell, Happé & Frith, 2000*). In this study people with AN were found to focus more on the details and showed negative interpretation bias when describing the movements of the triangles than healthy comparison participants even though no significant quantitative differences in task performance were observed (*Sedgewick et al., 2019*). This suggests that the present version of Ambiguous scenarios task could be a useful tool to assess both quantitative and qualitative differences in top-down cognitive processing of ambiguous, social stimuli.

In both exploratory factor analyses positive film clips loaded onto the same factor while negative film clip loaded onto another factor, suggesting the positive and negative film clips successfully evoked the intended mood and facial affect. This finding suggests that both measurement modalities produced similar results, but the correlation results suggest that this was only the case with some of the positive film clips. Although strong reactions were recorded and reported in response to the negative film clips, no significant correlations were observed between the two measures. Another recent study also using similar measures also found that while there was a positive correlation between facial affect and self-reported mood when viewing positive pictures, no significant correlation was observed when participants viewed negative pictures (*Höfling, Föhl & Alpers, 2020*). Similar lack of association between facial expressions and self-reported mood has been previously reported in response to negative film clips (*Wang, Marsella & Hawkins, 2008*; *Höfling, Föhl & Alpers, 2020*). These findings may reflect the fact that self-reported mood
and spontaneous facial expressions can provide different types of information about a person's emotional experience highlighting the need to assess both. This is particularly relevant in the field of mood and eating disorders where people have been documented to report experiencing emotions while suppressing their outward display (*Davies, Schmidt & Tchanturia, 2013*; *Reed, Sayette & Cohn, 2007*).

The above findings may suggest that self-reported mood ratings and facial expressions provide different information. Similar findings have previously been reported in other studies documenting that facial and physiological reactions, such as changes in skin conductance, in response to emotionally provoking film clips did not significantly correlate with self-reported emotional states (*e.g.*, *Fernández et al., 2012*; *Wilms & Oberfeld, 2018*; *Gabel et al., 2019*). Along the same lines, studies exploring emotion regulation have reported that instructing participants to down regulate their emotions through reappraisal results in reduction in self-reported mood, but not necessarily in facial or physiological reactions to emotionally provoking stimuli (*Lalot, Delplanque & Sander, 2014*; *Mohammed, Kosonogov & Lyusin, 2021*). Such findings have led suggestions that having an emotional experience is not the same as being aware of it and asking participants to rate or otherwise explain their current feelings would require emotional awareness and possibly some top-down interpretation (*Kim & Fesenmaier, 2015*; *Russell, 2003*). This seems to suggest that facial and physiological reactions are distinct from self-reported mood states and utilising both could enable simultaneous examination of both top-down and bottom-up emotion generation pathways. As everyday emotional experiences have been proposed to be a result of a mixture of top-down and bottom-up processes (*McRae et al., 2012*; *Ochsner et al., 2009*), such a paradigm could be even more naturalistic and have greater ecological validity. Further investigation of the mechanisms that underlie differences between self-reported mood ratings and facial expressions may be of interest to explore whether both emotional reactivity and interpretation biases could be examined simultaneously.

In the present study we did not find any significant correlations between interpretations of or reactions to emotional film clips and the clinical self-report measures. This is unexpected considering that there is a wealth of previous work indicating that people with depression and eating disorders show reduced facial reactivity to emotionally provoking stimuli (*Panaite, Whittington & Hindash, 2018*; *Davies et al., 2016*; *Dapelo et al., 2016*; *Leppanen et al., 2017*). Similarly, several studies have also documented that negative interpretation bias is a key feature in depression and social anxiety (*Everaert, Podina & Koster, 2017*; *Chen, Short & Kemps, 2020*) and it has recently also been documented in eating disorders (*Rowlands et al., 2020*; *Dapelo et al., 2016*). On one hand, this may be because performance on interpretation of the ambiguous films and reactivity to the emotionally provoking films bear no association with depression, anxiety or eating disorder psychopathology. On the other hand, this finding may also be driven by the fact that the present sample consisted of healthy young women, which naturally led to reduced variability in the clinical self-report measures. Further investigation of whether performance on these tasks is associated with relevant clinical measures among people with anorexia nervosa, depression or anxiety may be of interest to further shed light on whether these film stimuli target the relevant aspects of social-emotional processing in these illnesses.

**Table 5** Recommended film clips.

| Ambiguous scenarios task | | | Evoked emotions task | | |
|---|---|---|---|---|---|
| Category | Film code | Film name (min) | Category | Film code | Film name |
| Negatively valenced | Film 2 | Fast and Loose (0:00 –1:42) | Negative | Film 1 | In a heartbeat (0:12 –2:13) |
| | Film 6 | Pillars (6:12 –7:39) | | Film 2 | Invisible Strings (1:57 –2:30, 4:02 –5:37) |
| | Film 9 | Pregnant Pause (0:41 –2:14) | | Film 3 | Lifeboat (5:32 –8:33) |
| | Film 11 | Whenever You're Ready (1:47 – 3:30) | | Film 4 | Presentation (4:47 –7:10) |
| | Film 12 | Youth, full (0:35 –2:36) | | Film 5 | Pride and Pack –Pride of Lions (17:58 –20:09) |
| | Film 17 | The Proposal (0:10 –1:49) | | Film 6 | Work (0:18 –2:22) |
| Positively valenced | Film 7 | Bare (3:58 –5:52) | Neutral | Film 9 | Dreaming whilst Black (1:16 –3:32) |
| | Film 10 | The Liberty (5:45 –7:39) | | Film 10 | Ohio (5:36 –7:47) |
| | Film 13 | City Lights (3:57 –5:12) | | Film 11 | Reception (0:07 –2:24) |
| | Film 14 | Don't Be a Hero (7:46 –9:31) | | Film 12 | RPG (0:00 –2:10) |
| | Film 15 | Hello, Again (3:37 –5:16) | | Film 13 | Blessing in Disguise (3:14 –5:32) |
| | Film 16 | Palm Trees and Power Lines (3:01 –4:45) | | Film 14 | Reality 2.0: Catcalling (0:07 –2:50) |
| | | | Positive | Film 15 | Chinese Hi-Five (0:11 –2:17) |
| | Film 18 | So It Goes (3:28 –5:24) | | Film 16 | Hot mess (0:24 –2:49) |
| | | | | Film 17 | Russian Roulette (1:12 –3:46) |
| | | | | Film 18 | Standby (0:21 –2:26) |

**Notes.**

Min, minutes.

## Recommendations for use

The film clips we recommend for the Ambiguous scenarios and Evoked emotions tasks are listed below in Table 5. Based on our findings most of the negative film clips in the Ambiguous scenarios task appeared to yield primarily negative responses and we would, therefore, not recommend most of them. Instead, we recommend that some of the film clips initially categorised as neutral but were interpreted primarily as neutral or negative. These films would provide enough variance in participants responses to statistically examine differences between clinical and general populations without risking ceiling effects. Additionally, Film 17, which was originally categorised as positively valenced, may also be used to as a negatively valenced ambiguous film, particularly in terms of question 2, which asks participants to make predictions about how the situation might turn out. We would recommend the use of most of the film clips initially categorised as positively valenced with the exception of Films 17. Finally, based on our findings we cannot recommend any of the films clips evaluated here to be used as truly neutral ambiguous stimuli due to some degree of bias being observed in all films.

Most of the film clip used in the Evoked emotions task elicited the intended mood ratings and facial affect. Only two films, both of which were initially labelled as neutral (Films 7 and 8), should be excluded based on the present findings for producing unexpectedly positive and negative responses. If these films clips are used to study evoked facial affect, it is of note that there was substantial variability between in the facial affect data. Thus, large sample size may be needed to examine true differences between any clinical and healthy control

groups. Additionally, care should be taken during the video recording of participants' faces to ensure the recordings can be analysed accurately. For instance, lighting can impact any automated facial expression analysis software's ability find the face and participants should be instructed to not move too much or cover their faces by resting the heads in their hands.

## LIMITATIONS

The main limitation of this study was the small sample size. Although the sample size was sufficient for the ambiguity check, it may not have been sufficient for the exploratory factor analyses. Even though the two exploratory factor analyses overall produced similar results, the small sample size may go some way to explain the number of cross-loadings observed with the neutral film clips. Furthermore, due to the SARS-CoV-2 pandemic, most participants had to take part in the study remotely from home. This may have impacted data quality introducing variability in the quality of the video recordings of participants' facial expressions, which may have impacted the FaceReader analysis and thus the factor analysis. Furthermore, a recent study found that FaceReader's valence measure may not distinguish well between reactions to unpleasant and neutral stimuli, even when the difference could be established using facial electromyography (*Höfling, Föhl & Alpers, 2020*). The authors suggest that FaceReader may lack sensitivity in accurately detecting true negative facial expressions while simultaneously having negative bias while analysing neutral faces. This is relevant as in the present study the neutral film clips loaded across all factors in the factor analysis that used facial expression data, which was not the case in the factor analysis using mood ratings. Thus, greater accuracy of computerised facial affect recognition tools further examination of participants' facial affect under controlled laboratory conditions are needed.

Additionally, it is possible that the lack of association between facial affect and participants mood ratings was due to the use of FaceReader. We averaged the valence of participants facial expressions over the duration of the film clip. As most participants displayed neutral facial expressions most of the time, this approach led to only small differences in valence of facial affect between the film categories. Other studies have explored the use of dominant basic emotion rather than using averaged emotion intensity or valence scores, which has been found to be comparable to the manual Facial Action Coding System (FACS) (*Skiendziel, Rösch & Schultheiss, 2019*). However, such an approach would result in categorical data, which does not take other secondary emotions expressed during the film into consideration. Additionally, there is some uncertainty regarding whether FACS-based scoring would provide greater accuracy (*Limbrecht-Ecklundt et al., 2016*). Therefore, more work is needed to establish suitable methods to summarise the frame-by-frame facial expression data produced by computerised analysis tools, such as FaceReader.

Another limitation was that we used a convenience sample of young women aged 18 to 25 years. Thus, the results cannot be generalised beyond this population. To expand beyond this sample, further evaluation of the stimuli is needed with a more diverse and larger sample. Additionally, the sample size is modest, particularly for factor analyses.

Thus, further examination of the stimuli in the context of Ambiguous scenarios and Evoked emotions tasks is still needed.

Finally, although we did not find statistically significant carry-over effects in either task, some trend level effects were observed in the Evoked emotions task. It appeared that participants reactions to neutral and positive films clips may have been somewhat influenced by the films being preceded by other positive films. This suggests that increasing the inter-stimulus interval for a task assessing participants mood and evoked facial expressions beyond 200 ms may be required to eliminate any potential carry-over effects. It is possible that similar trend level carry-over effects were not seen in the Ambiguous scenarios task because participants were required to write answers to two questions after each film clip, which took longer than rating their mood and alertness. Thus, the ultimate inter-stimulus interval in the Ambiguous scenarios task extended beyond that on the one in the Evoked emotions task.

## CONCLUSIONS

This study aimed to evaluate two sets of 18 film clips to be used in computerised tasks assessing interpretation of ambiguous scenarios and evoked emotional responses. Altogether, 124 healthy female participants aged 18 –25 took part in the evaluation study. The participants completed two tasks in which they watched all film clips and responded to questions after each clip. During the Evoked emotions task participants facial expressions were also recorded. Due to the SARS-CoV-2 pandemic more than half of the participants completed the study remotely. The ambiguous film clips elicited the intended primarily neutral, positive, and negative interpretations while remaining ambiguous. However, three of the film clips were deemed unambiguous. Additionally, participants were generally more attuned to the negative cues suggesting that to preserve ambiguity more subtle negatively valenced film clips should be used to assess interpretation biases. Participants' evoked facial expressions and mood ratings in response to the film clips used in the Evoked emotions task were analysed with two exploratory factor analyses to identify film clips that elicited the intended reactions. The factor analyses revealed that the positive and negative emotionally provoking film clips formed their own factors, while there was substantial cross-loading with the neutral film clips when facial affect data was used. This could reflect reduced data quality due to most of the participants taking part remotely with webcams and internet connections of varying quality. Still, the findings show that a subset of the film clips evaluated in the present study could be used to assess interpretation biases and emotional reactions using a new, more ecologically valid set of stimuli within a clinical population.

### Funding

This research was funded by the Wellcome Trust [213578/Z/18/Z]. For the purpose of open access, the author has applied a CC BY public copyright licence to any Author Accepted Manuscript version arising from this submission. The research was further supported by MRC-MRF Fund [MR/R004595/1]. There was no additional external funding received for this study. The funders had no role in study design, data collection and analysis, decision to publish, or preparation of the manuscript.

### Grant Disclosures

The following grant information was disclosed by the authors:
The Wellcome Trust: 213578/Z/18/Z.
MRC-MRF Fund: MR/R004595/1.

### Competing Interests

The authors declare there are no competing interests.

### Author Contributions

- Jenni Leppanen conceived and designed the experiments, performed the experiments, analyzed the data, prepared figures and/or tables, authored or reviewed drafts of the article, and approved the final draft.
- Olivia Patsalos analyzed the data, prepared figures and/or tables, authored or reviewed drafts of the article, and approved the final draft.
- Sophie Surguladze analyzed the data, authored or reviewed drafts of the article, and approved the final draft.
- Jess Kerr-Gaffney analyzed the data, authored or reviewed drafts of the article, and approved the final draft.
- Steven Williams conceived and designed the experiments, authored or reviewed drafts of the article, and approved the final draft.
- Ketevan Tchanturia conceived and designed the experiments, authored or reviewed drafts of the article, and approved the final draft.

### Human Ethics

The following information was supplied relating to ethical approvals (i.e., approving body and any reference numbers):

King's College London Psychiatry, Nursing and Midwifery Research Ethics sub-committee (ref: HR-19/20-13004).

### Data Availability

The data is available at OSF: Leppanen, Jenni. 2022. "Development and Preliminary Validation of a Social-Emotional Processing Task Battery." OSF. February 1. doi: 10.17605/OSF.IO/AZR63.

## Supplemental Information

Supplemental information for this article can be found online at http://dx.doi.org/10.7717/peerj.14160#supplemental-information.

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
