# Peer review of "Evaluation of film stimuli for the assessment of social-emotional processing: a pilot study"

_PeerJ, doi:10.7717/peerj.14160_

## Round 0.1 · original submission · Minor Revisions

Both reviewers raise a number of points that will help to enhance the clarity of the manuscript and further improve its quality.

·

Basic reporting

This is an interesting study evaluating two sets of films to be used in the assessment of (1) interpretation bias and (2) evoked emotional responses. There is a need for more ecologically valid means to elicit both interpretation bias and emotional responses in experimental settings; thus, the study is relevant and adds to the literature, providing stimuli that can be used in future studies. The study is well conducted, and statistical analyses are sound. I do, however, think the article would benefit from rewording some bits to enhance clarity, before publication.
Abstract:
When I first read the abstract, I found it very packed with information and slightly confusing. When I re-read it after reading the whole article, it made a lot more sense. Thus, there is nothing wrong with the abstract per se, however, I do worry that people making a quick literature search might skim it, find it confusing, and abstain from reading the article in full, which would be a shame. I would suggest the authors to take a second look at it and perhaps eliminate some unnecessary wording (e.g., “The aim of the current study was to build on existing work and develop ….”, could be reduced to “The aim of the study was to develop…”, which is more concise and straightforward). Also, I would suggest, to find a way to separate the methods and findings of each film-set to make it more clear to a naïve reader. At the moment, for example, it is not obvious in the abstract why one needs to check the “level of ambiguity” of some of the videos, and why the fact that some videos were not ambiguous is relevant.
Introduction:
Typo: Line 63: I think it should read “negative interpretation bias” (currently: “negative interpretation, bias”)
Overall, the introduction reads well. I think using “top-down” and “bottom-up” emotional responses as a framework for the study is very useful, because it links the study of “interpretation bias” (which is involved in “top-down” emotional responses” and the study of “evoked emotions” (“bottom-up”). However, since the rest of the article focuses specifically on interpretation bias and evoked emotions, I would suggest the authors to clearly define both concepts (especially interpretation bias) and expand a bit on them. This is relevant, because the main aim is to evaluate if the sets of film clips evoke (1) interpretation biases (therefore, providing a stimulus that elicits an emotional response whilst being ambiguous enough to provide room for interpretation bias to occur), or (2) emotion (therefore, providing a stimulus that consistently elicits emotions of certain valence, without much room for interpretation). Thus, the reader needs to fully comprehend why the things that are being evaluated are relevant (e.g., why evaluating ambiguity is relevant in a set of clips that will be used in the evaluation of interpretation bias). At the moment, a reader that is not familiarized with the literature might not fully understand it. Moreover, there is little mention to “top-down” and “bottom-up” emotional responses in the rest of the article, so the excessive focus on these concepts during the introduction (particularly compared to the little focus on defining “interpretation bias” and “evoked emotions”) seems a bit inconsistent.

The aim of the study reads: “The aim was to evaluate 18 film clips to identify valid film clips to be used in an interpretation bias task and identify film clips, to evoke intended types of interpretations while still being ambiguous enough to be open for interpretation. We also evaluated a second set of 18 film clips to identify valid film stimuli to be used in an Evoked emotions task.”
I think “to be used in an interpretation bias task” and “to evoke intended types of interpretations while still being ambiguous enough to be open for interpretation” is a bit redundant. It is also confusing because the study is evaluating 2 sets of clips, and when reading that first part of the aim, the reader tends to think that the second sentence (“to evoke intended types of interpretations while still being ambiguous enough to be open for interpretation”) is referring to the second set of clips (the one for “evoked” emotions). But, as I understand, the authors are still taking about the first set of clips (the ones for measuring interpretation bias). I would suggest removing one of the sentences. In my opinion (but the authors may disagree), I would remove the first sentence (“to be used in an interpretation bias task”), because it places all the emphasis on the task, instead of emphasizing the need to develop a set of films suitable to elicit interpretation bias. Still, I would suggest rewording the second sentence a bit to not use the word “evoke” (which can be confusing since the second set of films is for “evoked” emotions), and to avoid repeating the word interpretation.
Likewise, for the second part of the aim (“We also evaluated a second set of 18 film clips to identify valid film stimuli to be used in an Evoked emotions task.”) I would suggest removing the emphasis from the task and to phrase it as identifying film stimuli that would evoke emotions.
In my opinion, the relevance of the study is in the development of stimuli that can consistently achieve something (the elicitation of interpretation bias in one case, and evoked “bottom up” emotions in the other case), regardless of the exact task being used to measure those bias/emotions. For example, one could use the set of “evoked emotions films” for a different task that requires the elicitation of emotions, knowing that there is evidence that they work. This is why I think the main focus should be on the assessment of the evidence that these films clips can provoke interpretation bias/ evoked emotions. Of course, you need a task to check if this is true, but I think the task is only a means to evaluate the behaviour. Putting all the emphasis on the tasks (e.g., the aim is to develop film clips to be used in task X) narrows the scope of the study. This is a comment, but not necessarily something to be changed in the article. I would be interested in hearing what the authors think about it. As I mentioned above, they may very well disagree with me.

Methods:
I understand that there is no space to describe the 36 film clips and it does make sense to just include the links in a supplementary table. However, I think it would be good to get a description of one video as an example for each set to get a general idea. Also, I was wondering how the authors qualitatively evaluated ambiguity in the films when selecting. Perhaps an example contrasting an “ambiguous film clip” (selected for the measurement of interpretation bias) with a film selected to elicit bottom-up emotions might help? In which way the two sets were different? (I assume the ones selected for evoking emotions were not ambiguous since they were supposed to consistently elicit an emotion valence without room for interpretation?)

Coding of interpretation bias task: I would suggest the authors to provide more details about the way the interpretation task was coded. Currently, the article states: “The written responses were coded as either positive, neutral, or negative on the scale from 1(positive) to -1 (negative), with 0 representing neutral responses.” As a reader, I was left wondering if there were any guidelines for this coding. It would be helpful to provide an example of what was coded as “positive”, “negative”, or “neutral”. Also, I did not understand if the codes were categorical (e.g., 1 or 0 or -1) or in a continuum (e.g., 1 or 0.8 or 0.5, etc). Please clarify.

Since the affective slider was used to assess participants’ mood in the evoked emotions task, I think it should be described in the measurements section

In the sentence “For example, if the answers in response to a given film clip were strongly associated with negative interpretations, this was taken to suggest that the film lacked adequate ambiguity, with the majority of participants offering only negative interpretations” I think the last portion (“the majority of participants offering only negative interpretations”) sounds like you are describing results, but it’s written in the methods section.

Results:
I may have missed it, but I could not find any description/rationale for the “number of words” results. There is a column on the table showing “number of words” results, but I don’t think the authors mention them in the article. What do they mean?

Not a criticism, but only a comment: I think is interesting and somewhat unexpected that interpretation bias (particularly negative) did not show correlation with clinical variables. One would have expected at least some correlation with depression for example. On the other hand, these are healthy participants, so depression scores were not high. It would be interesting to follow up in a case control study. If the films clips work, one would expect a negative bias in the clinical group, compared to the non-clinical group, providing further support for the use of this set of film clips.

Discussion:
I think the discussion reads well.

Experimental design

Design and methods are ok, other than the comments written in #1

Validity of the findings

Results are ok, other than the comments written in #1

Reviewer 2 ·

Basic reporting

To the editor
Peer Journal
RE: Evaluation of film stimuli for the assessment of social-emotional processing: A pilot study (#70500)

General comments:
The study "Evaluation of film stimuli for the assessment of social-emotional processing: A pilot study (#70500)" is an interesting study aiming to construct a data base of short film clips to assess two complimentary processes in the generation of emotional responses. Top-down emotion generation involving cognitive appraisal of situations, and bottom-up emotion generation, referring to reactions arising from exposure to emotionally provoking stimuli. The findings of this preliminary study are incomplete, in particular with respect to cognitive appraisal. Nonetheless, it is an important first step in the formation of a data base of film clips that would have the potential to differentiate the generation of emotional processes in health vs. pathology, and to assess whether adequate interventions will close the gap between health and pathology
The basic reporting is adequate in all aspects required. The experimental design is adequate and innovative. The statistical analysis is excellent. The findings are innovative, and their validity is robust and statistically sound. The conclusions are well stated, although some theoretical broadening, as shown later, can further improve the quality of the paper.
In conclusion, I recommend the acceptance of the article to Peer Journal following the following corrections:
Abstract
Delete the last sentence in the Discussion. It is not connected with the results of the study.
Methods
Participants
Page 10, line 147: The participants are not randomly selected, but rather a convenience sample. This should be mentioned in the limitations of the study. In addition, the sample is relatively small. How was the number of the partiicpants decided?
Page 10, line 154: I did not find the reference of the SCID-5 (First et al, 2015) in the reference list.
Film stimuli
Page 10, line 173: Why were the films chosen for this healthy cohort based on themes included in a previous study of people with anorexia nervosa?
Page 11, line 192: In the results section it seems that participants disclosed whether the saw previously a film included in the study, which was then not included in the analysis. But this should be described here, in the Methods section
Self-report Measures
Page 11, page 201: Why were the patients required to respond to an eating disorders questionnaire (EDE-Q)? I can understand the inclusion of questionnaires assessing the severity of anxiety and depression, bit why including the EDE-Q, a questionnaire assessing the severity of anorexia nervosa and bulimia nervosa, if the use of the SCID-5 criteria and the self-report information of the BMI, likely excluded participants with disordered eating from the study
Page 12, line 244: To readers unfamiliar with the affective slider and the automated facial affect analysis tool FaceReader, as this reviewer, please provide a brief description of these techniques.
Page 12, lines 265-266: From the way it is written here it is not clear whether there were no differences between participants completing the tasks in person and those who took part from home. This should be stated


Results
Page 17, line 413: evoked emotion task: I assume that the film clips s in the evoked emotion talks were not required to be ambiguous, as was in the ambiguous scenarios task, but this should be clarified
Discussion
Several theoretical and technical elaborations can be added in my opinion to the discussion to strengthen the study
1. Would longer film clips increase their ambiguity in the ambiguous scenarios task?
2. Could theoretically the same films clips be administered at the same time for both the ambiguous scenarios task and the evoked emotion task?
3. Elaborate on the differences between mood self-report and facial affect analysis in the evoked emotion task. From the findings of the study it seems that they are not similar
4. Elaborate on possible interactions between top down and bottom up emotion generation processes
Typos
There are numerous typos in the study. To mention just a few:
Page 8, line 107: " Furthermore, reduced in facial expression of emotions…"
Page 8, line 127: "These findings suggest…"
Page 18, line 482: ”This interactions was driven by a significant impact…."
Page 19, line 537: ”… anxiety and depression, as measured by HADS, depression
Page 21, line 598: ”This is particularly relevant in the field of mood….

Experimental design

Put in the basic reporting

Validity of the findings

Put in the basic reporting

Additional comments

No

Annotated reviews are not available for download in order to protect the identity of reviewers who chose to remain anonymous.

---

## Round 0.2 · accepted · Accept

Both reviewers are satisfied with your revision. Please make sure to consider the following points raised by one of the reviewers, eg, when uploading the final version of the manuscript or during the proof process:

Page 21, line 629: "Another factor at play" and not as is written (paly)
Page 24, line 676-677: Not a complete sentence
Page 25, line 711: In my opinion the authors should add: ".....among people with anorexia nervosa, depression or anxiety.....in these illnesses"
Page 26, lines 779-782: Delete. This material is not relevant for the present study
Page 26, lines 782-782: Delete, as it appears already in page 25, lines 745-747
Table 2 delete ":number of word in"" in headline, as this does not appear in the table

·

Basic reporting

No comments

Experimental design

No comments

Validity of the findings

No comments

Additional comments

The authors have addressed all my previous comments and suggestions satisfactorily, and I think now the article is ready for publication.

Reviewer 2 ·

Basic reporting

OK. The authors have made all the required revisions

Experimental design

OK

Validity of the findings

OK

Additional comments

NO